# Temporal nutrition analysis associates dietary regularity and quality with gut microbiome diversity: insights from the Food & You digital cohort

Rohan Singh [1], Daniel McDonald[2], Alejandra Rios Hernandez[2], Se Jin Song [3], Andrew Bartko [2,4], Rob Knight[2,3,5,6,7] & Marcel Salathé [1] ✉

The gut microbiota is profoundly influenced by dietary choices, with emerging evidence linking it to various health outcomes. Here, we investigate diet-microbiota associations using detailed temporal nutrition intake data captured through real-time food logging via a smartphone app and gut microbiota profiles from 16S rDNA sequencing in ~1,000 participants from a digital cohort on personalized nutrition ("Food & You" - clinicaltrials.gov NCT03848299). The primary outcome of the parental trial was to investigate post-meal glucose response variations between individuals in function of their individual factors such as diet, microbiome composition and lifestyle. Our analysis reaffirms that high-quality diets rich in vegetables, fruits, nuts, micronutrients, and favorable dietary indices like HEI (calculated both as standard HEI and daily HEI to capture day-to-day diet quality regularity) correlate with increased microbial diversity and improved stool quality, while fast food-rich diets show opposite effects. Regular consumption of beneficial food groups emerges as a key factor, with regularity in both food intake and diet quality sometimes showing stronger associations than average intake quantities. Machine learning analyses reveal strong bidirectional predictability between gut microbiota composition and dietary factors (ROC AUC up to ~0.85-0.9). These findings highlight the critical role of both diet quality and regularity in shaping gut microbiota, the importance of temporal nutrition tracking in offering insights for targeted nutritional strategies, and suggest that the gut microbiota can be used to estimate dietary indices.

The relationship between diet and the human microbiota has become a focal point in understanding human health and disease. The human microbiota, which includes trillions of microorganisms such as archaea, bacteria, viruses, fungi, and protists, plays a crucial role in metabolic, immune, and neurological systems of the host[1]. The composition and functionality of the gut microbiome are strongly influenced by diet, with diet quality and fiber intake being key determinants of microbial diversity and/or influencing the abundances of certain

[1]Digital Epidemiology Lab, School of Life Sciences, School of Computer and Communication Sciences, EPFL, Geneva, Switzerland. [2]Department of Pediatrics, University of California, San Diego, CA, USA. [3]Center for Microbiome Innovation, University of California San Diego, La Jolla, CA, USA. [4]Department of Bioengineering, University of California San Diego, La Jolla, CA, USA. [5]Halıcıoğlu Data Science Institute, University of California San Diego, La Jolla, CA, USA. [6]Shu Chien-Gene Lay Department of Bioengineering, University of California San Diego, La Jolla, CA, USA. [7]Department of Computer Science and Engineering, University of California San Diego, La Jolla, CA, USA. ✉e-mail: marcel.salathe@epfl.ch

phyla[2,3]. Dietary fibers, predominantly found in fruits, vegetables, whole grains, and legumes, have been known to be essential for a healthy microbiome[4,5]. These fibers are fermented by gut microbes, producing short-chain fatty acids (SCFAs) like acetate, propionate, and butyrate, known for their anti-inflammatory effects and ability to strengthen the gut barrier[5,6]. Hence, a diverse fiber-rich diet fosters a varied microbiota, enhancing the presence of beneficial bacteria and reducing harmful strains, thereby promoting better overall health[1,3]. In contrast, diets low in fiber and high in processed foods, such as "westernized" diets, contribute to a less diverse and less stable microbiota, increasing the risk of pathogenic bacteria proliferation and associated health issues like obesity, type II diabetes, and inflammatory bowel disease (IBD)[2,7]. The dynamic interaction between diet and the microbiota emphasizes the potential for dietary interventions to encourage beneficial bacterial growth, thereby improving health outcomes.

Recent studies have further deepened our understanding of how dietary components and patterns are linked with the gut microbiota and, consequently, host health. For instance, Asnicar et al.[8] have demonstrated, in a large cohort of >1000 participants, significant associations between the gut microbiota to food groups and habitual diets, as well as its connections to clinical and cardiometabolic health markers[8]. Furthermore, the Mediterranean diet, which is characterized by consumption of vegetables, fruits, vegetables, legumes, nuts and olive oil, has been associated with changes in the gut microbiota, and is linked to lower cardiometabolic disease risk[9,10]. On the other hand, research by Bowyer et al.[11] has shown that using dietary indices, such as the Healthy Eating Index (HEI), can effectively control for diet in gut microbiota studies, meaning that when compared with other dietary indices (Mediterranean Diet Score and Healthy Food Diversity index), HEI had the strongest association with the microbiota and explained the most variance, thereby providing a more standardized method to assess the impact of diet on microbiota composition[11]. In another study involving a large cohort of over 3000 Chinese participants, both short-term and long-term plant-based dietary patterns, assessed using the plant-based diet index (PDI), were observed to correlate with gut microbiota composition, wherein long-term patterns explained more of the microbiota variance and were associated with beneficial microbial changes and improved cardiometabolic health[12].

Despite advances in understanding diet-microbiome relationships, a critical dimension remains largely unexplored: the temporal consistency of dietary intake. Classically, studies investigating nutrition and health often use dietary assessment methods like food frequency questionnaires (FFQs) and multiple 24-h dietary recalls. While these methods provide valuable data and can capture general consumption frequency, they are subject to limitations such as recall bias and misreporting[13]. For example, a recent study analyzing doubly labeled water measurements found that over 50% of dietary reports in major national nutrition surveys contained implausible energy intakes, with systematic underreporting that increased with BMI[14]. Moreover, they typically represent snapshots or retrospective averages rather than continuous monitoring of day-to-day consumption variability as it occurs naturally.

Our study addresses this fundamental gap by leveraging real-world data (RWD), i.e., collected from participants in their natural environments rather than controlled research settings, from the digital cohort "Food & You," comprising 1013 participants across Switzerland. While all dietary assessment methods capture participants' actual eating behaviors, we use RWD to specifically denote our in-situ, real-time dietary logging approach that minimizes recall bias by capturing consumption at the moment it occurs, in contrast to retrospective methods that rely on participants' memory of past intake. Using the AI-assisted mobile application MyFoodRepo, we collected high-resolution nutritional data through real-time, in situ food logging[15],

thereby allowing us to extract temporal consumption habits as novel features for analysis.

In this work, we show that dietary regularity and quality are significantly associated with gut microbiota diversity. Our investigation into the interrelationship between the gut microbiota and various dietary and nutritional features reveal consistent patterns that align with findings from other geographical regions. Additionally, we explore microbiota relationships with a new temporal feature set: the regularity of food group consumption, characterized by the coefficient of variation (CV) of consumption quantity across tracking days. These findings not only reinforce the established diet-microbiota associations from previous studies but also underscore the potential of RWD in supporting the development of tailored dietary strategies to foster health-supporting microbiota.

## Results

### Diet quality and regularity influence gut alpha diversity

Alpha diversity, i.e., within sample diversity of microbial species, has previously been shown to be linked to dietary differences, particularly in terms of diet quality and fiber intake[3,11]. In our study, we derived two HEI-2020 measures for each participant: "HEI", a single score calculated from the average daily intake of each food group across all tracking days, and "daily HEI", the mean of HEI scores computed separately for each day, as a modified metric to capture day-to-day regularity in dietary quality. Our data exhibit the general expected trends as shown in Fig. 1A. Shannon entropy[16], a measure of microbial diversity within a sample, was observed to be positively correlated to HEI (Spearman $r = 0.22$), even more strongly to the daily HEI metric (Spearman $r = 0.27$), and to the consumption of fiber (g/day), vegetables-fruit content (g/day) and micronutrients such as potassium (mg/day), magnesium (mg/day), folate (μg/day) and iron (mg/day) (Fig. 1A). Similar correlation patterns were observed across other alpha diversity metrics including Pielou's evenness, Faith's PD, and observed features (Supplementary Fig. 1B). Conversely, unhealthy diet intakes such as fast foods (g/day) and salt (g/day) content displayed negative correlation to microbial alpha diversity (Fig. 1A). We also observed a positive correlation with age, with significant differences in diet quality and alpha diversity between younger and older age groups (Fig. 1E, H).

Analysis of participants grouped by median intakes of different food groups (g/day) revealed distinct patterns (Fig. 1B). Participants with above-median consumption of vegetables-fruits and fibers ($n = 142$) showed higher HEI diet quality, higher Shannon diversity, and lower BMI. In contrast, those with higher-than-median meat and fast food intake ($n = 144$) exhibited low HEI and Shannon diversity, with significantly higher BMI. Notably, participants consuming above-median levels of vegetables-fruits, meat, and fibers ($n = 89$) showed the highest Shannon diversity, along with very high HEI scores, older age, and lower BMI. Those with high fast food consumption alone ($n = 87$) demonstrated low HEI scores and Shannon diversity, and were predominantly younger.

Multiple regression analysis incorporating potential interactions revealed that HEI significantly predicted Shannon entropy ($\beta = 0.011$, $p = 0.035$, 95% CI [0.0008, 0.0213]) independent of age and gender, while accounting for confounding factors i.e., BMI, food quantity consumed (in g), hunger levels, and daily defecation frequency (for further details see Supplementary Table 2). The relationship between HEI and Shannon entropy remained consistent across age groups and genders (Fig. 1C). Notably, the regression model also revealed that BMI significantly influenced microbial diversity, where individuals with obesity ($\beta = -0.1891$, $p = 0.005$, 95% CI [−0.3208, −0.0528]) and individuals who are overweight ($\beta = -0.10$, $p = 0.011$, 95% CI [−0.1924, −0.0364]) both showed lower Shannon entropy compared to those with normal BMI. Additionally, it also indicated daily defecation frequency ($\beta = -0.137$, $p < 0.001$, 95% CI [−0.1833, −0.0929]) to be significantly negatively associated with microbial diversity. Furthermore,

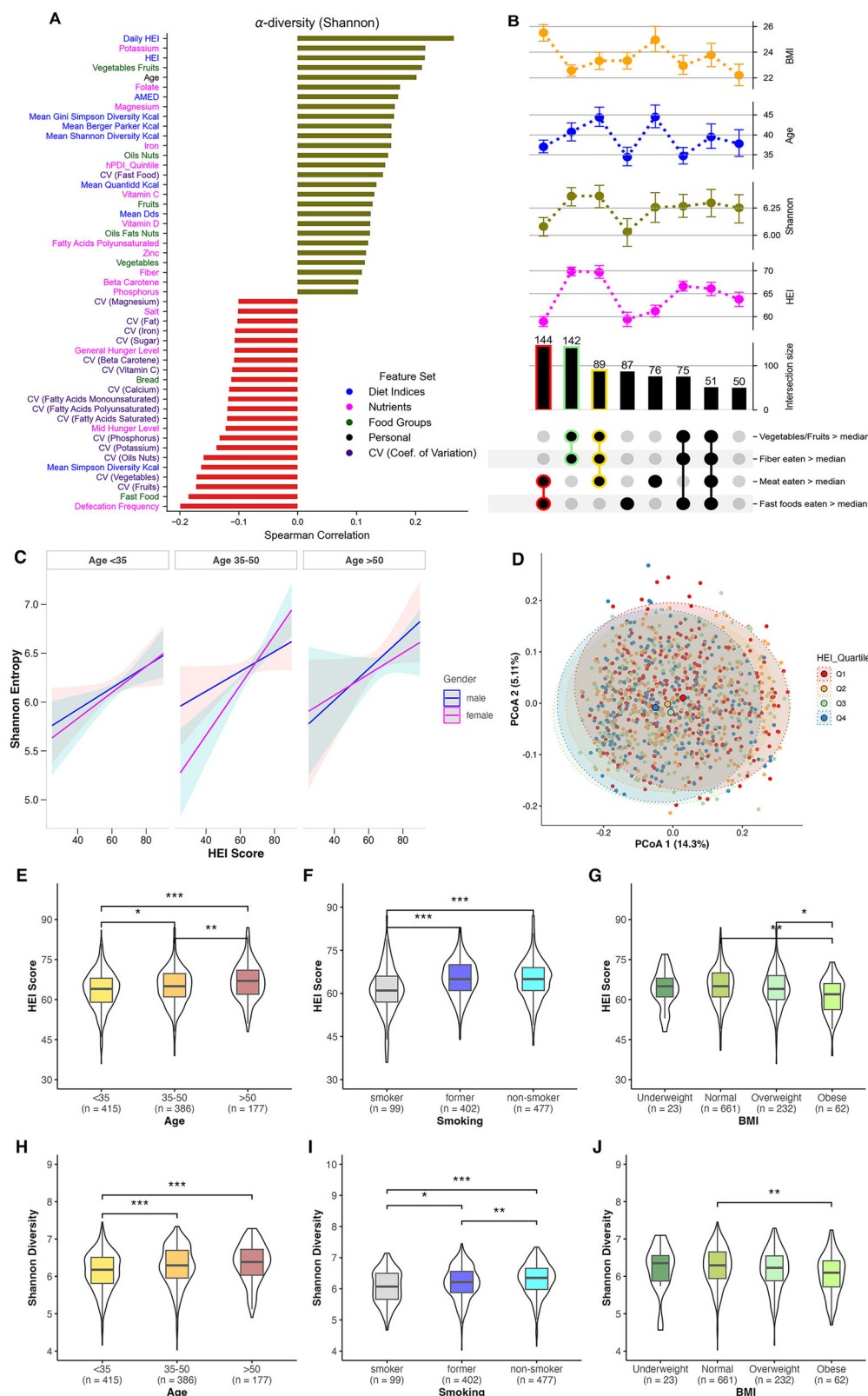

smoking status emerged as a significant factor, with both current smokers ($\beta = -0.1210$, $p = 0.0004$, 95% CI [−0.2888, −0.0692]) and former smokers ($\beta = -0.1789$, $p = 0.0016$, 95% CI [−0.1876, −0.0544]) showing reduced microbial diversity compared to non-smokers, while simultaneously having poor HEI (Fig. 1F, I).

Moreover, when we fit a model using daily HEI instead of the regular HEI score (with age and BMI as continuous variables and

without any interaction terms), daily HEI remained a highly significant predictor of Shannon entropy ($\beta = 0.019$, $p < 0.001$, 95% CI [0.0141, 0.0242])) (Supplementary Table 3), reflecting an even stronger effect than the standard HEI coefficient would have shown in that same model ($\beta = 0.013$, $p < 0.001$, 95% CI [0.0085, 0.0179]).

Since the real-time tracking via the MyFoodRepo app allowed us to collect temporal diet data across multiple days, we were able to also

**Fig. 1 | Alpha diversity comparison across physiological and dietary factors and demographic differences with respect to subsets of individuals differing by their median nutritional intake (g/day). A** Highlights correlations of different physiological and dietary factors with Shannon diversity. **B** The top section display point plots showing the mean values and variability of Shannon microbiome diversity, HEI, BMI, and age across different subsets of the population categorized based on their dietary habits: consumption of vegetables/fruits, fiber, meat, and fast foods relative to the median. Each dietary habit category is indicated with black dots for presence above the median. The numbers above each bar indicate the number of unique participants (*n*) in each dietary subset. Error bars in point plots represent the standard error of the mean. Notable subsets are additionally highlighted with a color outline, such as red for higher-than-median meat and fast food intake. **C** Represents the relationship between HEI scores and Shannon entropy across age groups and gender, as estimated by a multiple regression model adjusted for BMI, smoking, eaten quantity, general hunger level, and daily defecation frequency. Lines indicate the model-predicted mean values, with shaded areas representing 95% confidence intervals. **D** Principal Coordinates Analysis of beta diversity (unweighted Unifrac distances) and the variance explained by first two principal coordinates. Boxplots (**E**–**J**) highlight significant differences across age, BMI, and smoking with respect to HEI and Shannon alpha diversity. One-way ANOVA with post-hoc pairwise *t*-tests was used to assess differences in HEI scores (**E**–**G**), while Kruskal–Wallis tests with post-hoc Wilcoxon rank-sum tests were employed for Shannon diversity comparisons (**H**–**J**). Boxplots display the median (line), interquartile range (box) and whiskers extending to 1.5× IQR. All *p*-values were adjusted using the False Discovery Rate (FDR) method. Significance levels are denoted as * <0.05, ** <0.01, and *** <0.001. Abbreviations:- HEI Healthy Eating Index-2020, CV coefficient of variation.

calculate a metric that allows us to assess the temporal variability of consumption of different dietary components across participant's tracking days. This variability is expressed in terms of the CV, wherein higher values indicate higher irregularity of consumption. In Fig. 1A, we can see that the CV of different dietary features, such as fruits, vegetables, oil-nuts, mono- and poly-unsaturated fats as well as the aforementioned micronutrients were negatively correlated with gut microbiota diversity. Moreover, the absolute value of the correlation of these features were often higher for their variability than for their respective consumption quantities, e.g, $CV_{fruits}$ showed a Spearman correlation of $r = -0.18$, while fruits eaten showed $r = 0.12$. This indicates that high variability in consumption of these food groups can also be used as a metric, perhaps as a better metric for certain dietary features, to measure the impact of diet on Shannon diversity. This relationship between CV and gut Shannon diversity persisted even when controlling for total consumption amounts through stratified analyses (see Supplementary Section for detailed validation).

Additionally, we examined the extent of variance explained by dietary variables on the Shannon alpha diversity using multivariate regression models (Supplementary Fig. 1). We observed that the variance explained by macronutrients only (i.e., carbohydrates, protein, fat, fiber, alcohol) was about 5% while micronutrients explained a larger part of the variance, nearly 10%. Collectively, micronutrients, macronutrients, diet indices and food groups explained nearly 20% of the variance in the alpha diversity. Previous studies have suggested that diet accounts for approximately 5%–20% of the variance in the microbiome composition[17]. Correspondingly, principal coordinate analysis on the unweighted UniFrac distances (i.e., a distance metric to measure between sample based on phylogeny, i.e., beta diversity) revealed that about 20% of the variance were explained by the first two principal coordinates (shown in Fig. 1D), although cluster segregations were not very distinct based on either HEI quartiles. Although, considering two extremes of HEI quartiles (Q1 and Q4), the highest HEI quartile (Q4) was characterized by markedly higher consumption of vegetables and fruits. In contrast, the lowest HEI quartile (Q1) exhibited high consumption of fast foods and sweets+salty snacks+alcohol food groups (Supplementary Fig. 1C).

### Effect size of variables on microbial diversities
Furthermore, using the *evident* package[18], we computed the effect sizes (Cohen's f for multi-category variables and Cohen's d for binary variables) of different demographic and dietary features on different alpha diversities and unweighted UniFrac beta diversity, shown in Supplementary Fig. 2. Language, past antibiotics usage and microbiome sequencing batch showed large effect sizes on alpha diversities. In the case of Shannon diversity, HEI had a strong impact. Pairwise effect sizes were also computed on features that were already binary or by sub-setting the data to two extremes groups, typically representing the top and bottom quartiles within features that were not binary. In pairwise case, HEI Quartiles (Q1 vs Q4), age group (18–35 vs 65 + ), hPDI Quintiles and BMI categories (i.e., Normal vs Obese) showed large

effect sizes. In the beta diversity metric, however, only the past antibiotics treatment and menopausal state had a large effect.

### Microbial correlations with dietary features
Furthermore, to investigate which microbial species correlate well with the different food groups and macronutrients, we performed partial correlation analysis, keeping age and BMI as covariates (Fig. 2A). Most microbes that were positively correlated to HEI, also correlated well with fiber, vegetables, fruits and nuts content and micronutrients like potassium, and these microbes were predominantly from the taxonomic order *Lachnospirales*. Figure 2A shows several of these taxa, notably *Lachnospira (ASVs 55, 137), Eubacterium (ASVs 73, 222), Brotaphodocola (ASV 25), Alitiscatomonas (ASV 74),* and *Muricoprocola (ASV 228),* etc. A few of the taxa with high correlations with HEI were also linked to producers of SCFAs production, e.g., ASV 73 and 137. This makes sense, as fiber-rich diets increase SCFA production in the gut, primarily associated with butyrate-producing bacteria. These bacteria contribute to healthier outcomes due to the predominantly beneficial role of SCFAs. Interestingly, several non-*Lachnospirales* bacteria also correlated with HEI and related food features. These included ASVs from order *Oscillospirales*, like ASV 51 (*Dysosmobacter*), and from *Christensenellales* (ASV 100). Many taxa belonging to genus such as *Mediterraneibacter* (ASV 136) *Lawsonibacter* (ASV 134) were linked to higher meat consumption. While many other taxa showed positive correlation to fast food consumption and CV variables like $CV_{oil-nuts}$ and $CV_{Fruits}$ and were also negatively correlated with fiber and HEI index. These included taxa like ASV 386 (genus *Negativibacillus*), ASV 380 (genus *Merdibacter*), ASV 370 (genus *Acutalibacter*) and ASV 360 (genus *Thomasclavelia*), and many other ASVs from different taxonomic order like *Actinomycetales, Eryspilatotrichales, Lactobacillales,* etc.

ASV 73 (genus *Eubacterium_J*), prevalent in ~80% microbiome samples, was strongly associated with healthy eating intake, as it correlated with HEI, fiber, oil-nuts-seeds, vegetables and fruits. Here again, when comparing the top and bottom quartiles (Q4 and Q1, respectively) of HEI across individuals, we see a large difference where its median CLR abundance in the top quartile was much higher than its median CLR abundance in the bottom quartiles of these food groups (Fig. 2B). In the cases of meat, fast food, sweets+salty-snacks+alcohol and bread food groups, the median CLR abundance in the Q4 (top) quartiles of these food groups was lower than that of respective bottom quartiles. Another ASV with significant presence in people consuming higher quantities of these healthier food groups was ASV 100, while its presence was very low in people who consume more meat or fast foods (Fig. 2C). However, contrasting correlations were seen in the case of ASV 37 (genus *Dysosmobacter),* which had higher correlation with respect to meat consumption, and irregular consumptions of vegetables, fruits, oil-nuts. Its prevalence was ~90%, with its median CLR abundance in the Q1 quartile higher for healthier food groups (Fig. 2D). This indicates that these bacteria are common in poor diets, while they are less abundant in individuals consuming more vegetarian or healthier diets. Studies have indicated these bacteria to be linked to

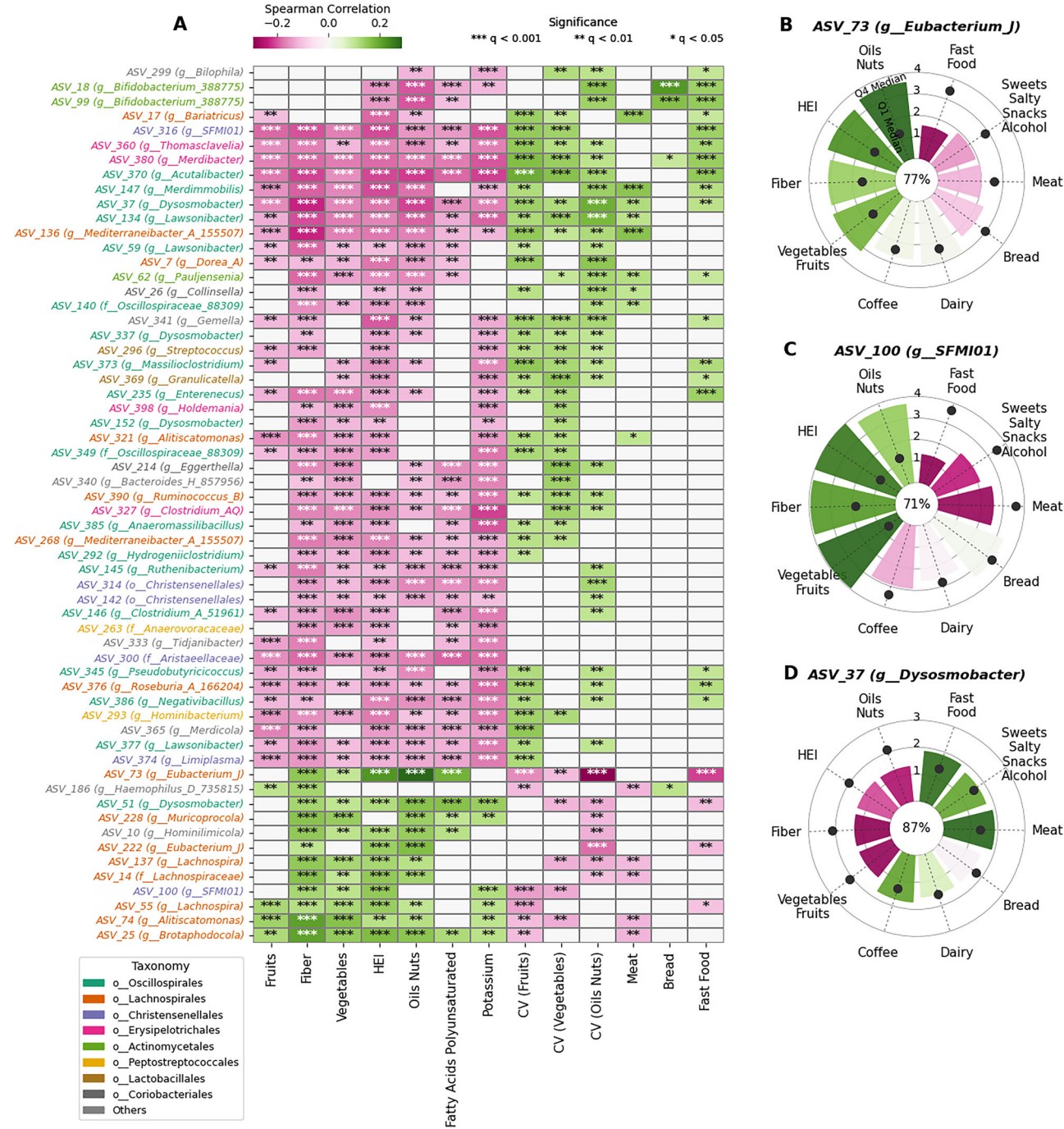

**Fig. 2 | Diet and microbiota taxa correlations. A** Partial Spearman correlation of significant microbes across different diet features. The significance of the correlation is indicated by a dot in each cell, which contains the adjusted *p*-values (FDR Bonferroni Hochberg correction). **B–D** Show radial plots highlighting the median CLR (Centered Log Ratio) of different microbes in the top (Q4) and bottom (Q1) quartiles of different food groups and the correlation with the respective food group. The radial bars show the Q4 median CLR abundance and the black dot corresponds to the Q1 median CLR abundance. The prevalence percentage of the microbe is shown in the center of the plot.

positive health outcomes in mice[19]. However, in our cohort, they correlated negatively with the HEI index.

**Differential taxa and log-ratio associations with diet**

We computed log ratios from the differentially abundant taxa (top 100), identified using BIRDMAn, across nutritional variables. Correlations of log ratios show strong complementary with their respective nutritional variables and other related variables, while contrasting patterns are seen with variables that are negatively associated with the corresponding nutritional variable, as depicted in Fig. 3E. For instance,

log ratio for HEI (Log Ratio$_{HEI}$) is strongly positively related to HEI ($r = 0.42$, $p < 1e$-$50$) while the same log ratio is inversely related to fast food consumption ($r = -0.27$, $p < 1e$-$16$), as shown in Fig. 3A, B. Similarly, dietary irregularity variables, i.e., CV, are inversely correlated to log ratios for various foods, like vegetables-fruits group, oil-nuts, fiber consumption and HEI (Fig. 3E). Comparing log ratio associations with other geographical cohorts[20] reveal similar patterns (Supplementary Fig. 1D, E). For example, microbial log ratios showed positive correlations with HEI across multiple countries, including the US ($r = 0.26$), UK ($r = 0.11$), and Mexico ($r = 0.097$). This was further supported by

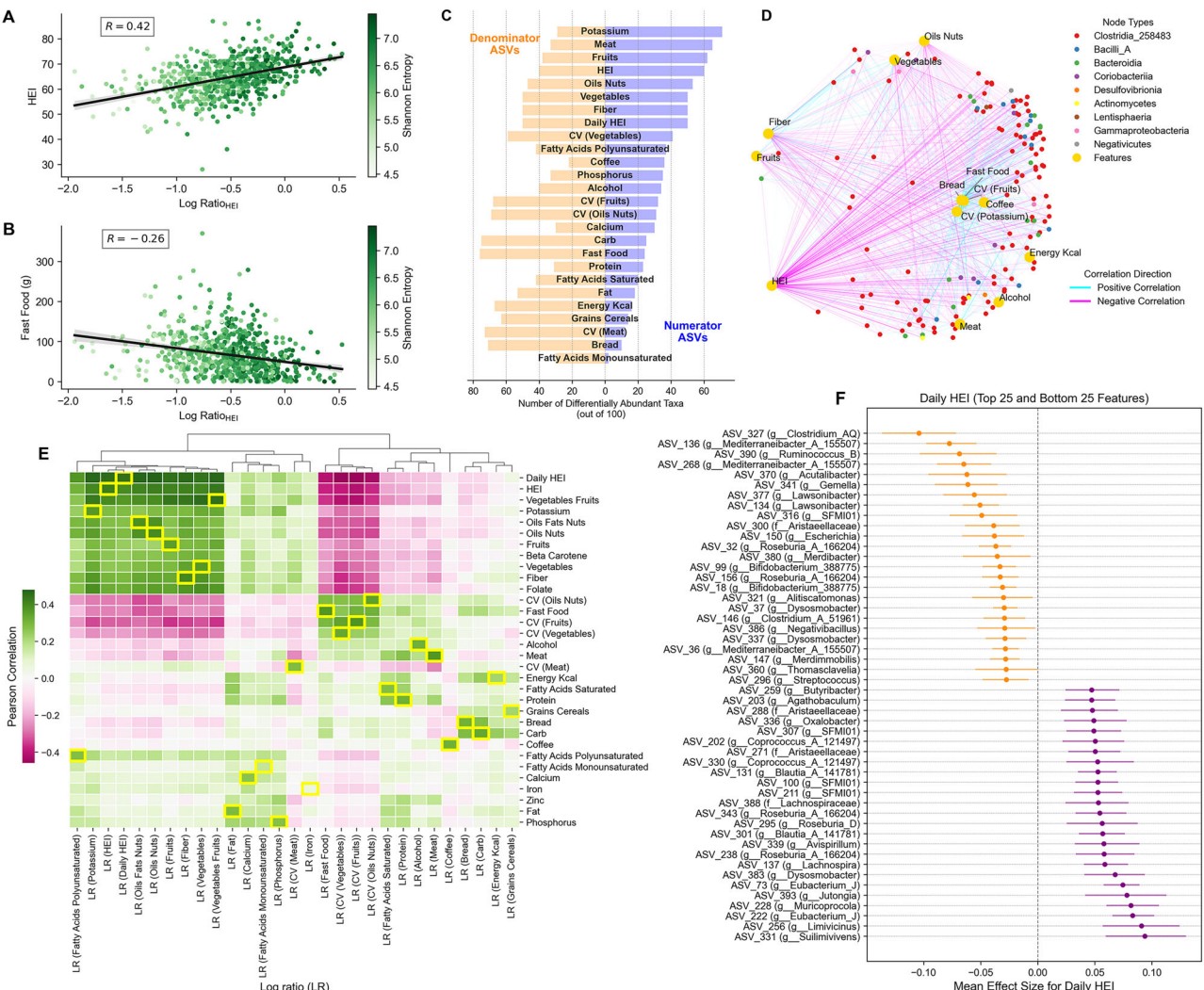

**Fig. 3 | Associations between microbial log ratios of differentially abundant microbes to dietary features.** Scatterplots in **A**, **B** highlight the Pearson correlation between the log ratio of HEI and either HEI or fast food intake. Each point represents a unique participant, colored by Shannon entropy. Black lines show the linear regression fit; shaded bands indicate 95% confidence intervals for the mean prediction. **C** Highlights the number of differentially abundant taxa (max 100, grouped as numerator ASVs and denominator ASVs) across different dietary features. Credible ASVs with positive mean effect sizes form numerator ASVs, while negative ones form denominator ASVs. Collectively, their abundances are used for computing log ratios. **D** Bipartite network visualization of correlations between dietary features and gut microbiome taxa at the class level. Yellow nodes represent dietary variable,s including food groups, HEI, and dietary variability (CV). Colored nodes represent different bacterial classes, with node colors corresponding to taxonomic classification. Edge colors indicate correlation direction (cyan: positive, magenta: negative), and edge thickness represents correlation strength. **E** Heatmap showing the correlations of log ratios of various dietary features to all other dietary features, wherein the correlations of the corresponding features are highlighted in yellow outline. **F** Shows the top 25 positively (purple) and bottom 25 negatively (orange) associated ASVs for daily HEI identified using BIRDMAn. Points indicate the mean effect size for each taxon; error bars represent the 95% highest density interval (HDI) for each taxon's association with HEI scores. Only taxa with credible intervals not crossing zero are shown.

positive associations with fiber intake ($r = 0.4$, $p < 2e-39$ in the "Food & You" cohort) that was also found in other populations (US: $r = 0.28$, UK: $r = 0.2$, Mexico: $r = 0.12$).

Figure 3D highlights associations between dietary components and differentially abundant gut microbial taxa at the class taxonomic level. The dietary components analyzed included HEI, fiber, fruits, vegetables, energy intake (kcal/day), food groups including meat, fast food, bread, coffee, and alcohol; CV for fruits and potassium. The gut bacterial community in the network was predominantly represented by *Clostridia*. Healthy dietary components (e.g., HEI, fiber, fruits, and vegetables) exhibited similar correlation patterns, showing positive associations (cyan edges) with certain bacterial taxa and negative associations (magenta edges) with others. HEI emerged as a key factor with numerous connections, reflecting its strong association with microbiome composition among differentially abundant taxa.

BIRDMAn analysis identified several taxa with strong positive and negative associations with daily HEI score (Fig. 3F). Among the taxa showing decreased log mean-abundance with increasing daily HEI scores, multiple taxa from the genus *Mediterraneibacter* (ASV 136, 268, and 36) showed negative associations, along with members of *Dysosmosisbacter* (ASV 37 and 337) and *Lawsonibacter* (ASV 377 and 134). These taxa exhibited lower abundance in individuals with higher scores. Conversely, taxa showing positive associations included several members of the *Lachnospiraceae* family and multiple taxa from the genera *Eubacterium_J* (ASV 222 and 73), *Butyribacter* (ASV 259) and *Coprococcus_A_121497* (ASVs 330, 202) were among the taxa most positively associated with daily HEI, potentially indicating their preference for dietary patterns associated with higher scores. Interestingly, *Roseburia_A_166204* had taxa that were both negatively (ASV 32 and 156) and positively (ASV 343 and 238) differentially abundant.

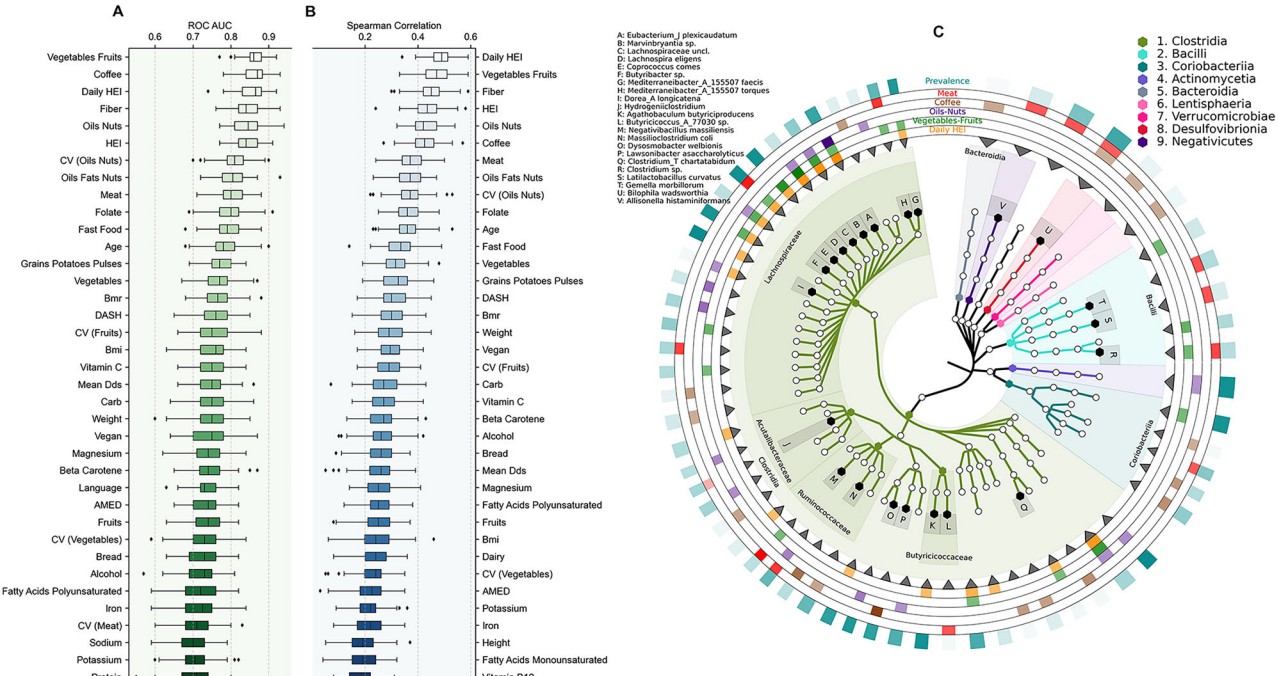

**Fig. 4 | XGBoost classifiers and regressor performance and feature importances. A** Shows the classifier performances (ROC AUC) on predicting the extreme quartiles (Q1 and Q4) of different physiological and dietary features using only the microbiome data ($n = 489$). **B** Shows the regressor performances (Spearman correlations) on predicting the values of different physiological and dietary features using only the microbiota data ($n = 978$). Each boxplot summarizes 100 independent train-test splits (80:20), with each point corresponding to a unique test set.

Boxplots display the median (line), interquartile range (box) and whiskers extending to 1.5× IQR. **C** Represents a phylogenetic tree highlighting the key microbial taxa that had the highest feature importance in different dietary classifiers (for daily HEI, vegetables-fruits, oil-nuts, coffee, and meat). Color intensity of the rings correspond to feature importance, except for the outer ring, which corresponds to the prevalence of the corresponding bacteria.

Differential abundance analyses were similarly performed for other dietary variables (Supplementary Fig. 4).

## Bidirectional prediction between diet and microbiota

Given previously reported associations of diet and lifestyle with health outcomes, we further inspected whether in our data, machine learning models could predict dietary and lifestyle factors using only the microbiota as feature inputs. We examined the distinguishability between extreme quartiles (Q1 vs Q4) for dietary features as a classification task using XGBoost classifier models. Figure 4A and Supplementary Fig. 3C show that consumptions of the food groups vegetable-fruit and coffee, as well as daily HEI regularity, are strongly predictable with AUROC and AUPRC values around 0.9. Similarly, we can also observe this predictive power for consumptions of fiber and the food group oil-nuts, as well as standard HEI diet index, which all show AUROC and AUPRC values around 0.85–0.9 (Fig. 4A and Supplementary Fig. 3C). Further, $CV_{oils-nuts}$, meat consumption, folate intake, fast food intake, grains-potatoes-pulses, vitamin C, and Dietary Approaches to Stop Hypertension (DASH) diet index were also strongly predictable, with ROC AUC values around 0.75–0.85. Age groups (18–35 and >50 years), BMI categories (normal vs obese) and linguistic regions of Switzerland (German vs Latin) were also predictable from microbiota features.

We then examined the regressor model performance using the entire dataset, not just the extreme quartiles used for the classifiers. The patterns observed in the regressor models closely mirror those seen in the classifier results (Fig. 4B). Specifically, the daily HEI and vegetable-fruit consumption show the highest Spearman correlation values, nearing 0.5. Of note, however, the predictive power of $CV_{fruits}$ was higher than that of fruits consumed, both in the case of classifier and regressor models. The same pattern is observed for the daily HEI and the standard HEI diet index, where the regularity metric outperforms the standard metric. We also reversed the prediction direction

to classify extreme quartiles of alpha diversity metrics using dietary and anthropometric features. Faith's PD showed the highest predictability (median ROC AUC ~0.75), followed by Shannon diversity and richness (number of observed microbiota features), shown in Supplementary Fig. 3A. In other words, microbial composition strongly predicts dietary patterns, which was also demonstrated in recent studies[8,21], while dietary patterns also demonstrate robust, albeit slightly lower, predictive power in the opposite direction.

Analysis of feature importance for the best performing dietary classifiers (HEI, vegetables-fruits, oil-nuts, coffee, and meat) revealed shared predictive taxa (Fig. 4C). Members of the *Lachnospiraceae* family, particularly *Eubacterium*, *Lachnospira*, and *Butyribacter*, were important predictors for HEI, vegetables-fruits and oil-nuts. For meat consumption, important predictors included *Mediterraneibacter* and *Coprococcus (Lachnospiraceae)*, along with *Blautia*, *Negativibacillus (Ruminococcaceae)*, several *Bacilli*, *Bilophila* and *Allisonella* genus. Coffee consumption was best predicted by *Lawsonibacter*, *Massilioclostridium* and members of *Coriobacteriia* and *Bacteroidia*.

When predicting alpha diversity metrics from dietary and lifestyle factors, daily HEI emerged as the strongest predictor across all four diversity metrics, followed by age and potassium intake (Supplementary Fig. 3B). Notably, several metrics of dietary variability (CV) ranked among the top predictive features, particularly CV of vegetables, HEI, and fruits. The prominence of these CV features in predicting alpha diversity provides independent support for our observation that both diet quality and its temporal consistency play key roles in shaping the gut microbiome composition.

## Stool quality linked to diet quality, diversity, and regularity

In the "Food & You" cohort, participants filled daily questionnaires, including optional stool quality reports, with 140 participants reporting for at least 5 days. Stool quality was categorized as normal, great,

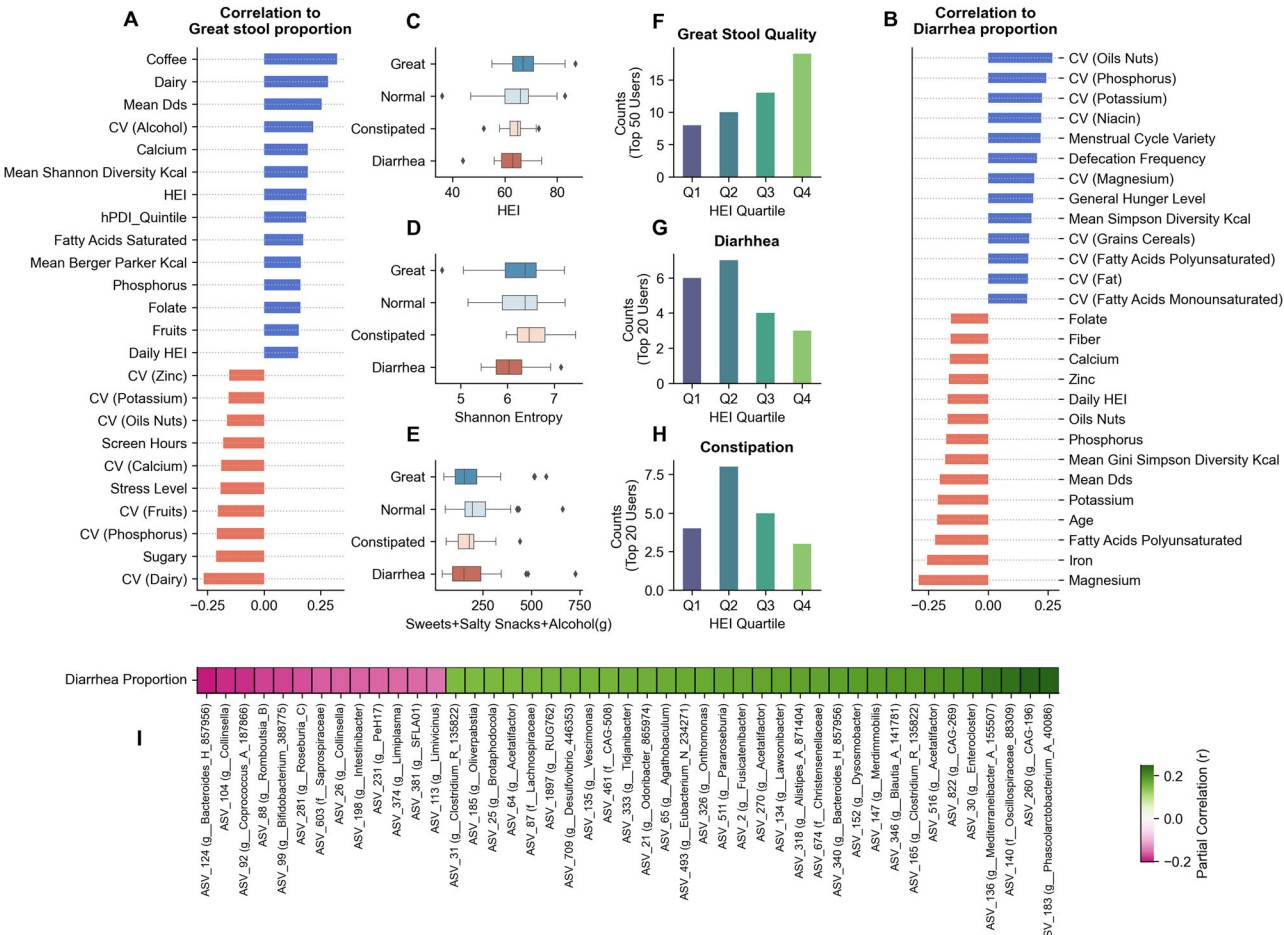

**Fig. 5 | Daywise stool quality analysis correlations. A, B** Represents the Spearman correlation (greater than ±0.15) of different factors to proportions of self-reported "great stool quality" proportion and diarrhea proportions, respectively. **C–E** Highlight the distributions of HEI, Shannon diversity, and sweet-salty snacks-alcohol consumption across top users grouped by stool quality: diarrhea (*n* = 20), constipated ($n = 17$), normal ($n = 61$), and great ($n = 43$). Boxplots display the median (line), interquartile range (box) and whiskers extending to 1.5× IQR. **F–H** Highlight the HEI quartile proportions for groups of individuals with different self-reported stool qualities. **I** Shows the heatmap highlighting the partial correlation of the most correlated bacteria (positive and negative) to the diarrhea proportion observed in users.

constipated, or diarrhea. We derived two features: great stool proportion, denoting the proportion of days reported as great, and diarrhea proportion, denoting the proportion of days reported as diarrhea.

Figure 5A shows the factors correlating with great-quality stool proportion. Coffee consumption showed the strongest positive correlation, followed by dairy, mean dietary diversity score (DDS), and $CV_{alcohol}$, with Spearman correlations ranging from $r = 0.2$ to $0.3$. Certain micronutrients (such as calcium, phosphorus, and folate), along with saturated fatty acids and diet indexes (HEI and dietary Shannon diversity), showed positive correlations between $r = 0.15$ and $r = 0.2$. Sugary foods and fast foods demonstrated negative correlations ($r = -0.22$ and $r = -0.15$, respectively). Several CV factors, including $CV_{dairy}$, $CV_{phosphorus}$, $CV_{fruits}$, $CV_{calcium}$, and $CV_{oils-nuts}$, negatively correlated with great stool quality. Lifestyle factors such as stress levels, screen hours, and sleeping problems also showed negative correlations. Notably, for many dietary features, the correlation of irregularity with great stool was stronger than that of the feature's quantity.

For diarrhea proportion (Fig. 5B), irregular intakes of different food groups and nutrients ($CV_{oils-nuts}$, $CV_{phosphorus}$, $CV_{potassium}$, $CV_{grains-cereals}$, $CV_{niacin}$) showed positive associations. Defecation frequency, menstrual cycle variety, and general hunger level also positively correlated with diarrhea proportion. Micronutrient intakes (for magnesium, iron, potassium, phosphorus, zinc, calcium) and polyunsaturated fatty acids showed negative correlations ($r = -0.15$ to

$-0.28$). Dietary indices, including mean DDS, HEI, and Gini Simpson Diversity scores, also negatively correlated with diarrhea proportions.

We classified users based on stool quality proportions: the top 50 users with the highest great quality as great, the top 20 users with the highest constipated and diarrhea proportions as constipated and diarrhea, respectively, and the remainder as normal. Users with great stool quality showed higher occurrences of high HEI quartiles Q4 and Q3 (Fig. 5F), while diarrhea users showed the opposite trend (Fig. 5G) and lower Shannon alpha diversity (Fig. 5D). Constipated users showed no linear trend with the HEI, with most users having either above-average (Q3) or poor (Q1) diet quality.

Partial correlation analysis, controlling for age and BMI, revealed associations between microbial taxa and diarrhea proportion (Fig. 5I). Genera including *Mediterraneibacter_A_155507* (ASV 136), *Phascolarctobacterium_A* (ASV 183), *Acetatifactor* (ASVs 516 and 270), *Bacteroides_H* (ASV 340), *Lawsonibacter (ASV 134), Dysosmobacter* (ASV_152), *Clostridium_Q_135822* (ASV 165 and 31) showed highest positive correlations. Conversely, ASVs corresponding to *Bifidobacterium_388775* (ASV 99), *Bacteroides_H* (ASV 124), *Coprococcus_A_187866* (ASV 92), and *Collinsella* (ASVs 104 and 26), demonstrated weak negative correlations (around $r = -0.2$).

## Discussion

The "Food & You" study was a digital nutrition cohort that collected RWD in situ from over 1000 healthy participants, including detailed

dietary data, glucose data, sleep and activity data, as well as a stool sample yielding microbiota data. Participants collected data with an AI-assisted app (MyFoodRepo) in real time over at least two weeks. Previous analyses have shown that the collected diet data is of good quality when compared with more traditional collection methods[15,22,23]. It is important to note that every single data point collected with the app was double checked (and corrected if necessary) by expert human annotators specifically engaged for the study, giving us confidence on the quality of the data. Real-time food logging through photo capture or barcode scanning, combined with automatic image recognition and subsequent human validation, potentially minimized recall bias by eliminating reliance on retrospective reporting, resulting in high completion rates (>90% over two weeks of tracking)[15]. While the study period overlapped with the COVID-19 pandemic, including periods of lockdown in Switzerland, which could potentially influence dietary patterns and food access, our statistical analyses did not detect significant year-to-year variations in energy intake (one-way ANOVA: $F = 0.97$, $p = 0.43$).

Here, we focus on the microbiota and its association with diet and anthropometric data, further strengthening the link between diet and gut microbiota. While confirming many associations reported in other studies and geographies, the highly detailed diet data from the "Food & You" study—particularly its temporal resolution—allowed us to investigate the link between microbiota and diet regularity. We find that in many ways, the regularity of intake of certain food groups is as important as the quantity of those groups, sometimes even more so. This insight adds a new dimension to our understanding of the diet-microbiome relationship and may have implications for personalized nutrition strategies.

The results reinforce evidence of a correlation between microbial alpha diversity, as measured by Shannon entropy, and diet quality, particularly the HEI. These associations extend to fiber, vegetable-fruit content, and micronutrients, including potassium, magnesium, folate, and iron, reflecting dietary fiber and micronutrients' essential role in maintaining a diverse gut microbiome[3,24]. Our data revealed demographic associations with dietary habits and gut diversity: While older participants consumed healthier foods and exhibited higher microbial diversity (Fig. 1E, H), multiple regression analysis showed that HEI predicted Shannon entropy independent of age (Fig. 1C), aligning with findings that bacterial diversity increases with age until 60–80 years[25]. It is important to note that while the estimated effect of HEI on Shannon diversity is 0.019 per unit, its cumulative impact could be considerable. Given that the HEI scale ranges from 0 to 100, an improvement in dietary quality of 10–20 points may correspond to an increase in Shannon diversity of roughly 0.19–0.38 units. In this way, even moderate enhancements in diet quality could potentially drive biologically relevant shifts in the gut microbiome. Conversely, unhealthy dietary practices, including high consumption of fast foods, sugar, alcohol and salt, were negatively correlated with microbial alpha diversity (Fig. 1A) and are linked to lower microbial diversity and health issues like obesity and IBD[7].

The regularity of intake of certain food groups and nutrients was often associated with gut diversity and stool quality, with regularity sometimes showing stronger associations than quantity. Irregularity in consuming fruits, vegetables, oil-nuts, and unsaturated fats negatively correlated with Shannon diversity (Fig. 1A). Machine learning classifiers using microbiota features predicted strongly different food groups and nutrition intake regularity (Fig. 4A, B), while irregularities also showed a pronounced association with stool quality (Fig. 5A, B). Moreover, the reverse prediction of microbiome diversity from dietary factors showed modest predictability, wherein HEI and CV metrics were among the top predictors, supporting the importance of dietary regularity (which has not been analyzed in past microbiome studies). While conventionally, HEI is calculated as a single score across days[26,27], we find that our daily HEI score is more strongly associated with gut

diversity than the standard HEI score, highlighting the importance of day-to-day dietary quality regularity.

Mechanistically, the stronger association of dietary regularity (as measured by the CV) with gut microbial diversity may be understood from an ecological perspective. Regular daily nutrient delivery (i.e., a low CV) likely promotes a stable gut environment where microbial populations can flourish, leading to higher diversity and evenness. In contrast, high variability in food intake (i.e., a high CV) suggests intermittent nutrient availability that may disrupt microbial homeostasis, causing fluctuations in community structure and potentially reducing overall diversity. A recent study demonstrated that the pattern and specificity of substrate (dietary fiber) utilization can significantly affect evenness and richness, independent of the absolute amount provided[28]. Our findings similarly suggest that while absolute intake levels offer a measure of overall nutrient exposure, the temporal pattern—captured by the CV—adds a valuable dimension that reflects how consistently the microbiota is supported. Future research should further explore these dynamics to clarify under which circumstances dietary regularity may be particularly critical for maintaining a robust and diverse gut microbial community.

Previous epidemiological studies using FFQs demonstrated the importance of food intake regularity on positive health outcomes, e.g., for fruits and vegetables[29], which have subsequently informed public health guidelines, such as the recommendation to eat five portions of fruit and vegetables every day[30]. While FFQs are valuable and indeed assess frequency, regularity cannot be fully ascertained with this instrument. In contrast, the data collected in the "Food & You" study can provide good estimates of regularity, but the study population is smaller and less diverse than that of previous epidemiological studies. The MyFoodRepo tool, at the time of the study, had documented limitations including portion size estimation errors and challenges with composed beverage classification[23]. While we employed trained nutrition annotators to verify entries, some systematic measurement errors may remain. These limitations could affect our "eaten quantity" variable, which associates with microbiome diversity in our models, and may introduce some noise into our day-to-day variability estimates. However, the validation study noted that despite these measurement errors, the tool demonstrates good overall agreement with the reference method (i.e., weighted food diaries) and is well-suited for identifying dietary patterns and eating habits[23]. The post-AI human verification step applied in our study provides reasonable confidence in our dietary data. The consistency of our findings with established diet-microbiome relationships provides further support to the overall data quality of the study. Additionally, the single time-point collection of microbiome samples limits our ability to establish causal relationships between dietary patterns and gut microbiota composition.

Key dietary and lifestyle factors also influence stool quality. Coffee and dairy intake, along with higher dietary diversity scores, were positively correlated with better stool quality, while sugary foods, fast foods, stress, sedentary lifestyle, and sleep issues showed negative correlations. For diarrhea, irregular intakes of oils, nuts, grains, and cereals were positively associated, whereas micronutrients like magnesium, iron, and zinc, along with polyunsaturated fatty acids and higher dietary diversity scores were negatively correlated. Our classification of users into great stool, diarrhea, constipated, and normal groups highlights that those with great stool quality consume healthier diets, as indicated by higher HEI quartiles, whereas participants with diarrhea have lower dietary diversity and quality. Sleeping problems were associated with reduced stool quality and increased diarrhea, aligning with recent research showing sleep's effects on gut microbiota[31]. While our daily stool quality assessments relied on participants' self-reports using predefined categories (e.g., great for ideal consistency and frequency, normal for average consistency and frequency) rather than standardized clinical measures like the Bristol Stool Form Scale, the observed consistent associations with both

dietary patterns and specific microbial taxa suggest these subjective measures captured meaningful information about digestive health that complements our more objective measures of diet quality and microbiome composition.

In conclusion, our study demonstrates the critical impact of diet on gut microbiome diversity and composition. Diets rich in fiber, vegetables, and fruits were associated with a more diverse and beneficial gut microbiome, while high variation in intake and diets high in processed foods showed reduced microbial diversity. These findings emphasize that both the quality and regularity of dietary intake may be crucial for maintaining a healthy gut microbiome. Ultimately, this study emphasizes the need for a more nuanced approach to nutritional recommendations that considers not only what we eat, but also how consistently we consume beneficial foods.

## Methods

### "Food & You" study design and dietary data collection

The "Food & You" study, a digital nutrition cohort completed by 1014 participants, was conducted in Switzerland from 2018 to 2023[15]. The objective of the study was to gather detailed and precise nutritional data along with continuous glucose monitoring, physical activity and sleep tracking for a period of 2–4 weeks. Food intake by participants was digitally mediated wherein participants recorded their dietary intake using the AI-assisted MyFoodRepo app made by the Digital Epidemiology Lab, monitored their glucose with continuous glucose monitors, and tracked physical activity and sleep through wearable devices or surveys. Additionally, the study collected a one-time stool sample for analyzing gut microbiota and included a standardized breakfast to assess dietary effects. The study protocol was approved by the Geneva Ethics Commission (approval number: 2017-02124) and registered with both the Swiss Federal Office of Public Health (SNCTP000002833) and ClinicalTrials.gov (NCT03848299). The full study protocol, including detailed methodology on data collection, food logging procedures, and nutritional assessment techniques, is described in Hérritier et al.[15]. The primary outcome was continuous glucose monitoring, aimed at assessing interindividual variability in glycemic responses to diet. Secondary outcomes included physical activity, sleep, and gut microbiome composition. Informed consent was obtained from all participants for the use of their data and samples in this study.

During the tracking phase of the study, participants documented their daily, regular food intake under real-world conditions, i.e., devoid of the need for clinical or other personal visits. This was facilitated using the MyFoodRepo mobile app, which allows for meal logging via photo captures, barcode scans, or manual entry. When photos are used, the application employs a food recognition AI to identify the items consumed and provide nutritional data. In the "Food & You" study, all data entry methods were validated by trained human annotators.

From this dietary data, various recognized dietary indices, such as the Healthy Eating Index (HEI-2020), alternate Mediterranean diet (aMED), plant based diet index (PDI) and DASH were derived. For the HEI-2020, we computed two complementary measures: (1) a single HEI score based on the average daily intake of each food group across all tracking days, and (2) a "daily HEI" calculated separately for each day and then averaged per participant to capture day-to-day dietary quality. Food items were categorized into predefined food groups, including vegetables, fruits, meat, dairy, grains_cereals, oils_nuts, sugary foods, and others. Fast food consumption specifically refers to commercially prepared foods typically associated with quick-service restaurants, including items such as hot dogs, hamburgers, pizza, and similar ready-to-eat convenience foods. Daily nutritional intake was measured in standard units: g/day for macronutrients and food groups, mg/day or mcg/day for micronutrients depending on typical dietary reference values, with vitamin A in International Units (IU) and vitamin E in Tocopherol Activity Equivalents.

During the tracking phase, participants also provided daily self-assessments of their stool quality through optional questionnaires. Participants were instructed to categorize their stool quality for each day according to specific definitions provided in the study materials: "great" (passing stool of their ideal consistency at their ideal frequency), "normal" (passing stool of their average consistency at their average frequency), "constipated" (infrequent and/or hard to pass stool), or "diarrhea" (loose and watery stool). For analysis, we calculated the proportion of days reported in each category for participants who completed stool quality questionnaires for at least 5 days ($n = 140$), deriving metrics including "great stool proportion" and "diarrhea proportion" to quantify the frequency of these stool qualities during the tracking period.

### Microbiome data processing

Participants in the "Food and You" study provided a single stool sample during their tracking phase, which were sequenced at Microsynth AG (Switzerland). Microbial DNA was extracted using the protocol described in Héritier et al.[15]. The V4 region of the 16S rRNA gene was amplified using 515 F and 806 R primers and sequenced via two-step Nextera PCR libraries. Microbiome preprocessing utilized QIIME 2 (version 2024.2)[32]. Demultiplexed single-end reads were denoised using Deblur (version 2024.2)[33] to construct amplicon sequence variants (ASVs), which were trimmed to 150 bp and filtered against the Greengenes2 database (version 2024.09)[34] for taxonomic classification. Samples were rarefied to 15k reads, retaining 992 samples for analysis. Diversity analyses included unweighted UniFrac distances[35] and various alpha diversity metrics (Faith's phylogenetic distance, Pielou's evenness, observed features, and Shannon diversity, as highlighted in Supplementary Table 1) computed using q2-diversity (version 2024.2).

### Principal coordinate analysis

Beta diversity was assessed using the unweighted UniFrac distance matrix, which was generated using QIIME 2. The distance matrix was imported into R using the "ape" library[36] and converted into a dissimilarity object. Subsequently, Principal Coordinates Analysis was performed on this matrix to visualize the microbial community structure. The first two principal coordinates were plotted to illustrate the variation between the samples.

### Differential abundance analysis using BIRDMAn

To identify taxa associated with dietary patterns, we performed differential abundance analysis using BIRDMAn (Bayesian Inference of Relative Distribution across Multiple taxonomic Assignments)[37]. The analysis was performed using raw count data from the feature table, filtered to include only features with a minimum prevalence of 250 samples (~25% of samples). A negative binomial linear mixed effects model was implemented through BIRDMAn, incorporating subject-specific random effects and using the first taxon as the reference for the log-ratio transformation. The model was fitted using variational inference with 500 posterior draws. Taxa were considered significantly associated if their highest density interval (HDI) did not overlap zero, indicating a credible difference in abundance between the extreme quartiles. The effect size for each taxon was calculated as the mean of the posterior distribution. For visualization, the top and bottom 25 taxa with the strongest credible associations were selected based on their mean effect sizes, and their distributions were plotted along with their respective HDIs. Complete credible differential abundance results for all nutritional variables and sequences are available in Supplementary Data 1.

### Microbiota log ratio calculation

The top 100 credibly differential taxa identified by BIRDMAn were used to calculate log ratios for each nutritional variable. Only taxa with

credible associations (defined as those whose HDI did not cross zero) were included in the analysis. These taxa were divided into two groups based on their mean association with the nutritional variable: those with positive associations formed the numerator group, while those with negative associations formed the denominator group. For each sample, counts for taxa in each group (numerator and denominator) were summed. To avoid undefined values, samples with zero counts in either the numerator or denominator were removed; only samples with counts in both the numerator, denominator and with non-null metadata were retained. For each remaining sample, the log ratio was calculated as the base-10 logarithm of the ratio between the summed counts of numerator taxa and denominator taxa. For all nutritional variables, the total sample number in creating the log ratios remained close to the total sample count, with the exception of mono-unsaturated fatty acids, for which this was reduced to about 940 samples. The relationship between these log ratios and the nutritional variables was then assessed using Pearson correlation. The counts for the numerator and denominator ASVs for different dietary variables are shown in Fig. 3C.

### Coefficient of variation calculation
Tracking days with total energy intake below 1000 kcal were excluded from the analysis. To ensure reliable assessment of temporal dietary variations, participants with 5 or fewer days of remaining tracking days were excluded. For each remaining participant, we calculated the day-to-day variability in nutrient/food-group intake using (CV = standard deviation/mean × 100%). After these quality control steps, 978 participants were retained for subsequent analyses.

### Diet diversity metrics calculation
In addition to calculating microbiota alpha diversities, we also calculated diet diversities to quantify the richness and variety of participants' daily diet, presented in Supplementary Table 1. These metrics evaluate dietary diversity by assessing kilocalorie distribution, food choice predictability, dominant consumed food categories, and nutritional variety, providing insights into both food variety and their nutritional impact. This selection of dietary diversity metrics was based on Hanley-Cook et al.'s (2023) systematic review[38], which discusses various applied approaches to measuring dietary diversity and quality in nutrition research. Finally, to summarize the dietary diversity for each participant, we computed the average of their daily dietary diversity metrics across all days.

### Correlation analysis
Spearman partial correlations between microbial taxa and various health-related variables were computed, adjusting for age and BMI as covariates, with significance determined through FDR correction (threshold = 0.1). Subsequent analyses extended to multiple variables, filtering to retain the top 50 microbes with the most substantial non-null associations across different dietary and health metrics.

### Machine learning analysis of diet-microbiota relationships
We employed XGBoost (version 2.0.3) models to investigate bidirectional relationships between gut microbiota composition and dietary/lifestyle factors. In the first approach, we used bacterial taxa collapsed at the species level to predict dietary and lifestyle patterns. For classification tasks, we distinguished between extreme quartiles (Q1 and Q4) of dietary and lifestyle variables, while regression tasks utilized the complete data range. Models were implemented with hyperparameters optimized for microbiome data (n_estimators=1000, max_depth=6, colsample_bytree=0.8, subsample=0.4, alpha=0.1, learning_rate=0.005 for classification; n_estimators=1000, learning_rate=0.05 for regression).

In the second approach, we reversed the prediction direction to classify quartiles of microbial alpha diversity metrics (Faith PD,

Shannon entropy, observed features, and Pielou's evenness) using dietary and lifestyle features. Input features included macronutrients, micronutrients, food groups, dietary indices, consumption variability metrics, and personal characteristics.

For both approaches, we performed 100 independent iterations using different random seeds for 80:20 train-test data splits. Model performance was evaluated using the area under the receiver operating characteristic curve (ROC AUC) for classification tasks and Spearman correlation for regression tasks. For variables with mean values below 0.001, we applied log10 transformation with a small constant (1e-6). The relative contribution of predictive features was assessed based on their accumulated gain in the tree splits across all iterations.

Detailed model performance metrics (AUROC and AUPRC scores for classifiers and Spearman correlations for regressors) across all 100 iterations for each dietary index and variable are provided in Supplementary Data 2. Feature importance scores for all taxa (at their putative species-level classifications) contributing to the classifier predictions are also provided in Supplementary Data 2.

### Reporting summary
Further information on research design is available in the Nature Portfolio Reporting Summary linked to this article.

## Data availability
16S rRNA gene sequencing data from this study are publicly available in the European Nucleotide Archive (ENA) under accession number PRJEB85942 and in Qiita under study ID 15880. Metadata containing clinical, demographic, and nutritional variables cannot be deposited publicly due to participant privacy and ethical restrictions. Access to this metadata can be requested by contacting the corresponding author, subject to institutional ethical compliance.

## Code availability
The analysis code used to generate the results is publicly available on GitHub (https://github.com/digitalepidemiologylab/dietary-consistency-and-quality-associated-with-gut-microbiota-diversity-paper), and archived at Zenodo (https://doi.org/10.5281/zenodo.16779103).

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

## Acknowledgements

We would once again like to express our deep gratitude to the participants of the "Food & You" cohort, and everyone involved in its development and completion. We thank Marouane Toumi at EPFL for sharing the diet diversity metrics that enriched our analysis. This work was supported by grants to MS of the Kristian Gerhard Jebsen Foundation, and through support to RS from the EPFLglobaLeaders programme, funded from the European Union's Horizon 2020 research and innovation programme under the Marie Skłodowska-Curie grant agreement No 945363. The funders had no role in the design or execution of this study, in the analyses and interpretation of the data, or in the decision to submit results. RK and DM are supported by NIH DP1AT010885, Danone Nutricia Research 191330, and the Foundation for Founders 3002919. AB, SS, and AR are supported by Danone Nutricia Research 191330. The dataset from THDMI was generated through support from Danone Nutricia Research and the Center for Microbiome Innovation.

## Author contributions

R.S.: Conceptualization, data analysis, methodology, visualization, software, investigation, writing—original draft. M.S. and D.M.: Study design, supervision, writing—review and editing. A.R.H., S.J.S., A.B., and R.K.: Writing—review and editing

## Competing interests

R.K. is a scientific advisory board member, and consultant for BiomeSense, Inc., has equity and receives income. He is a scientific advisory board member and has equity in GenCirq. He is a consultant for DayTwo and receives income. He has equity in and acts as a consultant for Cybele. He is a co-founder of Biota, Inc., and has equity. He is a cofounder of Micronoma and has equity and is a scientific advisory board member. The terms of these arrangements have been reviewed and approved by the University of California, San Diego in accordance with its conflict of interest policies. AB is a founder of Guilden Corporation and is an equity owner. The terms of these arrangements have been reviewed and approved by the University of California, San Diego in accordance with its conflict of interest policies. The remaining authors declare no competing interests.
