## [Transparent Peer Review file · Nature Communications]

Temporal nutrition analysis associates dietary consistency and quality with gut microbiome diversity: Insights from the Food & You digital cohort

Corresponding Author: Dr Marcel Salathé

Version 0:

Reviewer comments:

Reviewer #2

(Remarks to the Author)

The authors have adequately addressed the concerns raised in my review.

(Remarks on code availability)

Reviewer #4

(Remarks to the Author)

The authors consistently mention “Real world data” – however all dietary data is real world – food diaries record data as you go – this is point also raised by previous reviewer and not addressed adequately.

The authors mix food and nutrients throughout – for example the explanation of the calculation of the CV calls out nutrient but in fact the data is predominantly foods – this is just one of many examples.

Lack of fundamental understanding of dietary data – there are recommendations on how to calculate the HEI – and they just disregard them. No units presented anywhere for the dietary data.

The explanation of the CV is odd – which seems to indicate that its all about consistency as opposed to diet quality – which is at odds with the literature. Furthermore, the CV is a measure of the spread of the data – whereas if they wanted to look at consistency it should be ICC.

From a dietary data perspective, classifying individuals into low v high quartiles of intake is not progress anymore.

This is a very confused manuscript making a lot of claims that are not backed up. Fundamentally, it is hard to see the novelty from the diet perspective.

With regards the previous reviewers comments – I find that some are not adequately addressed. I have highlighted some examples below.

With regards the comments from previous reviewer:

b. Line 90: Real-time in situ food logging sounds very much like electronic dietary records, which have been used in nutrition epidemiology previously. This framing around the collection method being substantially better than other dietary data is not well supported by the references provided.

the answer is inadequate and there is still claims that this collection method is substantially better – and not backed up

h. Line 613: I'm concerned that the authors are overselling the novelty and accuracy of this tool. The paper cited for the validation of the tool lists significant limitations in the inability of the tool to classify liquids and beverages, and overestimation of portion sizes. This is important because the quantity eaten is a variable included in the models presented in supplementary table 1 and is significantly associated with the microbiome alpha diversity. It would be much more honest to present this tool along with its limitations rather than to frame it as superior to other tools available for research. Expert human annotators are also mentioned, but no explanation as to what makes these people experts is provided.

The answer falls short on the concerns for the tool in assessing diet.

D. Appropriate use of statistics:

a. Line 390: I have some concerns about the use of CV for HEI. In the instructions provided by the NIH for how to use HEI when there are multiple days of dietary records, researchers are instructed to use all of the days to calculate the HEI score, rather than averaging the HEI calculated for each day. I worry that this per-day calculation of HEI is using the index in a way it was never intended to be used.

The answer doesn't address adequately the concerns raised – at the least the authors could calculate the HEI as it was attended to be calculated.

(Remarks on code availability)

Version 1:

Reviewer comments:

Reviewer #4

(Remarks to the Author)

Abstract states "Fast food" – need a definition for this – as it can mean many things.

Words such as "temporal nutrition" are meaningless – do the authors mean temporal food intake?

Energy under reporting and over reporting needs to be addressed. How did the authors deal with under reporters. This is standard and very important practice in nutrition studies.

How did the authors deal with zero inflated dietary data – which will be even more of an issue if daily recording is done.

I highly recommend that the authors reconsider the wording that they are using. Diet consistency has a very distinct meaning in clinical nutrition and refers to the consistency of the food.

<https://www.sciencedirect.com/science/article/abs/pii/S0197457216300180#:~:text=Similar to the line spread,different food in dysphagia diet.>

Could the authors comment on the recent paper where that restricted dietary options do not consistently induce convergence of the faecal microbiota across individuals, emphasising the personalised response of the microbiota to dietary modulation. Furthermore, effect sizes were small. <https://egastroenterology.bmj.com/content/3/2/e100161>

The authors should acknowledge the well know definitions of ubiquitously consumed foods as consumed daily by almost everyone and episodically consumed foods as foods consumed in a less regular fashion. This will reflect what the authors are capturing in CV. For more information on these and how they are dealt with see in assessment of intake see:

<https://epi.grants.cancer.gov/diet/usualintakes/method.html#:~:text=The premise of the NCI,both parts of the model.>

The authors should acknowledge this and examine if they are seeing anything more than this in their analysis of CV and also their daily assessment of HEI – what happens if they correct for episodically consumed foods?

(Remarks on code availability)

na

Reviewers' Comments

Reviewer #4 (Remarks to the Author):

The authors consistently mention “Real world data” – however all dietary data is real world – food diaries record data as you go – this is point also raised by previous reviewer and not addressed adequately.

Answer: We appreciate the comment by the reviewer. Our intention of using this term is to align with the current usage, meaning observational data collected in day-to-day settings, and generally in real time. Some dietary data sources like questionnaires are often filled in after eating events, which has the problem of recall error. In contrast, our study leverages in-situ, real-time logging of dietary intake via a smartphone app, with photo documentation and human annotator validation. This approach captures diet at the moment of consumption, reducing recall error and providing a higher-fidelity representation of actual intake.

To address this concern and provide greater clarity, we have added the following sentence to the Introduction section: *"While all dietary assessment methods capture participants' actual eating behaviors, we use "real-world data" to specifically denote our in-situ, real-time dietary logging approach that minimizes recall bias by capturing consumption at the moment it occurs, in contrast to retrospective methods that rely on participants' memory of past intake."* This addition acknowledges that other methods also capture real eating behaviors while clarifying our specific technical use of the term "real-world data" and the unique advantages of our temporal approach.

The authors mix food and nutrients throughout – for example the explanation of the calculation of the CV calls out nutrient but in fact the data is predominantly foods – this is just one of many examples.

Answer: We thank the reviewer for pointing out the mix between food- and nutrient-level language. We have revised the manuscript throughout to clearly distinguish logged food-group measures from nutrients in relevant sections and have maintained a consistent notation style, thereby reducing any remaining ambiguity for the reader.

Lack of fundamental understanding of dietary data – there are recommendations on how to calculate the HEI – and they just disregard them. No units presented anywhere for the dietary data.

Answer: While we understand the frustration this may cause the reader, we have now further clarified this in the text. Regarding HEI calculation, we have recalculated the HEI following official HEI-2020 scoring standards and updated our methodology description accordingly. To clarify our approach, we have distinguished between the standard HEI (calculated from average daily intake across tracking days) and what we now term "Daily HEI" (score calculated separately for each day then averaged per participant). While the Daily HEI approach is non-conventional, it captures day-to-day dietary quality variation and shows meaningful and good associations with the gut microbiota that also reflect dietary consistency patterns. For the missing units, we apologize for this omission and have now added unit specifications throughout the manuscript. We have included a statement in the Methods section indicating that nutritional data were quantified using standard dietary assessment units: g/day for macronutrients and food groups, and mg/day or mcg/day for micronutrients.

The explanation of the CV is odd – which seems to indicate that its all about consistency as opposed to diet quality – which is at odds with the literature. Furthermore, the CV is a measure of the spread of the data – whereas if they wanted to look at consistency it should be ICC. From a dietary data perspective, classifying individuals into low v high quartiles of intake is not progress anymore.

Answer: We used CV to capture each participant’s day-to-day dietary variability, i.e., how much an individual’s intake fluctuates around their own mean, because it provides a person-specific metric that can be directly correlated with microbiome diversity. By contrast, ICC quantifies reliability at the cohort level (partitioning between- versus within-subject variance) and does not yield per-participant values for downstream analyses.

In our results, average intake (e.g., grams of fruits per day) and CV (variability in those grams across days) both associate with Shannon diversity: higher average intake correlates positively ($r = +0.12$), while greater CV correlates negatively ($r = -0.18$). Importantly, this use of CV does not undermine the role of overall diet quality, as HEI and other diet-quality metrics show even stronger associations with microbial diversity. In other words, “how steadily” foods are consumed appears to add explanatory power beyond quantity alone.

For completeness, we note that in a separate study (“Minimum Days Estimation for Reliable Dietary Intake Information: Findings from a Digital Cohort,” Singh et al. 2024; doi: 10.1101/2024.08.29.24312779 - now accepted at European Journal of Clinical Nutrition) we did employ ICC to determine the minimum number of days needed for reliable nutrient estimates at the cohort level. However, that ICC-based approach addresses group-level reproducibility rather than individual-level variability, so we chose CV here to rank participants by their personal intake consistency.

We use extreme quartiles (Q1 versus Q4) to evaluate how easily the microbiome can distinguish very low versus very high intake in classification tasks. However, we also ran XGBoost regressions on the full continuous dataset to predict intake values directly.

This is a very confused manuscript making a lot of claims that are not backed up. Fundamentally, it is hard to see the novelty from the diet perspective.

Answer: We respectfully disagree with the reviewer’s assessment. Our study provides novelty in several aspects: (1) the use of real-time, AI-assisted dietary tracking providing higher temporal resolution than traditional methods, (2) analysis of day-to-day dietary variability (CV) as an individual-level predictor of microbiome diversity, complementary to average diet quality measures, (3) bidirectional machine learning predictions between microbiome and dietary patterns, achieving high accuracy (AUC ~0.85-0.9), and (4) integration of daily stool quality reports with dietary patterns. While diet-microbiome associations are established, our analysis of temporal consistency, combined with predictive modeling and stool quality data, represents a meaningful advance in understanding how dietary patterns shape gut microbiota composition.

With regards the previous reviewers comments – I find that some are not adequately addressed. I have highlighted some examples below.

With regards the comments from previous reviewer:

b. Line 90: Real-time in situ food logging sounds very much like electronic dietary records, which have been used in nutrition epidemiology previously. This framing around the collection method being substantially better than other dietary data is not well supported by the references provided.

the answer is inadequate and there is still claims that this collection method is substantially better – and not backed up

Answer: We have revised the manuscript (particularly in the discussion and introduction sections) to remove these claims and provide more accurate descriptions of what our method enables.

h. Line 613: I'm concerned that the authors are overselling the novelty and accuracy of this tool. The paper cited for the validation of the tool lists significant limitations in the inability of the tool to classify liquids and beverages, and overestimation of portion sizes. This is important because the quantity eaten is a variable included in the models presented in supplementary table 1 and is significantly associated with the microbiome alpha diversity. It would be much more honest to present this tool along with its limitations rather than to frame it as superior to other tools available for research. Expert human annotators are also mentioned, but no explanation as to what makes these people experts is provided.

The answer falls short on the concerns for the tool in assessing diet.

Answer: We have substantially revised our discussion section to more thoroughly address the MyFoodRepo tool's limitations. The updated text now specifically acknowledges the quantified measurement errors from the validation study, including portion size estimation errors and composed beverage classification challenges. We discuss how these limitations may specifically impact our study variables, including the 'eaten quantity' variable in our models and potential effects on day-to-day variability estimates. While we maintain that human annotation helped mitigate some errors and that the tool remains suitable for dietary pattern identification as noted in the validation study (Zuppinger et al 2022), we now present a more balanced assessment that acknowledges both the tool's capabilities and its measurement limitations, allowing readers to interpret our findings with appropriate context.

D. Appropriate use of statistics:

a. Line 390: I have some concerns about the use of CV for HEI. In the instructions provided by the NIH for how to use HEI when there are multiple days of dietary records, researchers are instructed to use all of the days to calculate the HEI score, rather than averaging the HEI calculated for each day. I worry that this per-day calculation of HEI is using the index in a way it was never intended to be used.

The answer doesn't address adequately the concerns raised – at the least the authors could calculate the HEI as it was intended to be calculated.

Answer: We thank the reviewer for this important methodological aspect. In response to this feedback, we have substantially revised our analysis to address both the traditional HEI application and our research objectives regarding dietary consistency.

What we have done in the revised manuscript:

- 1. Added standard HEI calculation:** We now present results using the conventional HEI approach (aggregating all dietary data across tracking days to calculate a single HEI score)

per participant) as recommended by NIH guidelines. These new results still demonstrate that traditional HEI significantly associated to gut diversity ($\beta = 0.011$, $p = 0.034$).

2. **Retained daily HEI as a complementary approach:** We have relabeled our day-to-day approach as "daily HEI" and present it as an additional analysis specifically designed to capture dietary quality consistency. This approach revealed stronger associations with gut microbiome diversity ($\beta = 0.019$, $p < 0.001$) compared to traditional HEI.
3. **Provided clear methodological justification:** We explicitly acknowledge in the discussion that daily HEI represents a non-conventional application but argue it serves a distinct research purpose. As Kirkpatrick et al. (2018) noted, single-day HEI calculations may be justified when there is clear biological rationale linking daily diet quality variability to health outcomes.

Our dual approach allows us to demonstrate that: (1) traditional HEI predicts gut microbiome diversity as expected, validating our findings within established frameworks, and (2) day-to-day consistency in dietary quality (captured by daily HEI) shows even stronger associations, suggesting the gut microbiome may be particularly sensitive to dietary pattern regularity. This finding has important implications for dietary recommendations and would not have been detectable using only the standard aggregated approach.

Reviewers' Comments

Reviewer #4 (Remarks to the Author):

Abstract states "Fast food" – need a definition for this – as it can mean many things.

Answer: We thank the reviewer for this suggestion. We have added a clear definition of fast food in the Methods section.

Words such as "temporal nutrition" are meaningless – do the authors mean temporal food intake?

Answer: We thank the reviewer for pointing out this ambiguity. We have revised the abstract to clarify our meaning by specifying "temporal nutrition intake data" and "temporal nutrition tracking" to emphasize that we are referring to the time-resolved collection of dietary intake information over multiple days.

Energy under reporting and over reporting needs to be addressed. How did the authors deal with under reporters. This is standard and very important practice in nutrition studies. How did the authors deal with zero inflated dietary data – which will be even more of an issue if daily recording is done.

Answer: To address concerns about energy misreporting, we first compared our participants' dietary intakes against MenuCH, the Swiss national food composition database and dietary reference standard, as detailed in our previous validation study (Heritier et al 2022). This comparison revealed no major systematic differences in energy or macronutrient reporting patterns, suggesting that substantial under or overreporting was not a significant issue in our study population.

Additionally, we addressed energy misreporting through several approaches already described in our manuscript: (1) we excluded tracking days with implausible energy intakes <1,000 kcal from analysis, (2) all dietary entries were validated by trained human annotators, and (3) we acknowledge potential measurement errors in our Discussion section while noting that our validation study showed good overall agreement with reference methods. Regarding zero-inflated data, this is not a concern for our aggregated food group-level analysis, rather than individual foods, and it is normal and expected for participants to have zero consumption of certain food groups on given days, representing natural dietary variation rather than a statistical artifact requiring correction.

I highly recommend that the authors reconsider the wording that they are using. Diet consistency has a very distinct meaning in clinical nutrition and refers to the consistency of the food.

[https://www.sciencedirect.com/science/article/abs/pii/S0197457216300180#:~:text=Similar to the line spread,different food in dysphagia diet.](https://www.sciencedirect.com/science/article/abs/pii/S0197457216300180#:~:text=Similar%20to%20the%20line%20spread,different%20food%20in%20dysphagia%20diet.)

Answer: We agree and have replaced "diet consistency" with "dietary regularity" throughout the manuscript to avoid confusion with the clinical terminology for food texture modifications.

Could the authors comment on the recent paper where that restricted dietary options do not consistently induce convergence of the faecal microbiota across individuals, emphasising the

personalised response of the microbiota to dietary modulation. Furthermore, effect sizes were small. <https://egastroenterology.bmj.com/content/3/2/e100161>

Answer: We thank the reviewer for pointing to Vermeulen et al. That study tests dietary monotony (three items) rather than the temporal regularity of diverse, high-quality intake we analyze. Its observation of reduced diversity and dysbiotic shifts under perfectly standardized but nutritionally narrow intake supports our interpretation: regularity benefits the microbiome only alongside quality and diversity. Vermeulen thus serves as a negative control for substrate diversity, while our real-world data show that regularity plus quality associates with higher diversity. Both studies also highlight inter-individual responses, supporting personalized nutrition.

The authors should acknowledge the well know definitions of ubiquitously consumed foods as consumed daily by almost everyone and episodically consumed foods as foods consumed in a less regular fashion. This will reflect what the authors are capturing in CV. For more information on these and how they are dealt with see in assessment of intake see: <https://epi.grants.cancer.gov/diet/usualintakes/method.html#:~:text=The premise of the NCI,both parts of the model>. The authors should acknowledge this and examine if they are seeing anything more than this in their analysis of CV and also their daily assessment of HEI – what happens if they correct for episodically consumed foods?

Answer: Our CV metric may partly reflect the classic distinction between ubiquitously consumed and episodically consumed components as formalized by the NCI usual-intake framework, which models usual intake as $P(\text{consumption}) \times \text{consumption-day amount}$ with a two-part model for episodic items. That framework is designed for 24-hour recall data to estimate population usual-intake distributions and address measurement error, sometimes including an FFQ as a covariate. Our real-time, multi-day logging targets within-person regularity rather than usual-intake estimation, so we did not fit NCI models. Further analysis could be interesting, but would be beyond the scope of this paper.